randomised controlled trial; knowledge-based competency; digital training; capacity building; problem-solving intervention; adolescent mental health; India

**Corresponding author:**
Vikram Patel;
Email: vikram_patel@hms.harvard.edu

# Developing knowledge-based psychotherapeutic competencies in non-specialist providers: A pre-post study with a nested randomised controlled trial of a coach-supported versus self-guided digital training course for a problem-solving psychological intervention in India

Sonal Mathur[1] ⓘ, Helen A. Weiss[2], Melissa Neuman[2], Baptiste Leurent[3], Andy P. Field[4], Tejaswi Shetty[1], James E. J.[1], Pooja Nair[1], Rhea Mathews[1], Kanika Malik[5], Daniel Michelson[4,6] and Vikram Patel[7,8] ⓘ

[1]Sangath, New Delhi, India; [2]Medical Research Council International Statistics and Epidemiology Group, Faculty of Epidemiology and Population Health, London School of Hygiene and Tropical Medicine, London, UK; [3]Department of Statistical Science, University College London, London, UK; [4]School of Psychology, University of Sussex, Brighton, UK; [5]Jindal School of Psychology and Counselling, O.P. Jindal Global University, Sonipat, India; [6]Department of Child & Adolescent Psychiatry, Institute of Psychiatry, Psychology and Neuroscience, King's College London, London, UK; [7]Department of Global Health and Social Medicine, Harvard Medical School, Boston, MA, USA and [8]Department of Global Health and Population, Harvard T.H. Chan School of Public Health, Boston, MA, USA

## Abstract

We evaluated a digital learning programme for non-specialists to develop knowledge-based competencies in a problem-solving intervention for adolescents to examine the overall impact of training on knowledge-based competencies among learners; and to compare the effects of two training conditions (self-guided digital training with or without coaching) in a nested parallel, two-arm, individually randomised controlled trial. Eligible participants were 18 or older; fluent in Hindi or English; able to access digital training; and had no prior experience of delivering structured psychotherapies. 277 participants were enrolled from 31 March 2022 to 19 June 2022 of which 230 (83%) completed the study. There was a significant increase in competency score from pre-training (Mean = 7.01, SD = 3.29) to post-training (Mean = 8.88, SD = 3.80), 6 weeks after the pre-training assessment. Knowledge competency scores showed larger increase among participants randomised to the coaching arm (AMD = 1.09, 95% CI 0.26–1.92, $p$ = 0.01) with an effect size ($d$) of 0.33 (95% CI 0.08–0.58). More participants completed training in the coaching arm ($n$ = 96, 69.6%) compared to the self-guided training arm ($n$ = 56, 40.3%). In conclusion, a coach-supported remote digital training intervention is associated with enhanced participation by learners and increased psychotherapeutic knowledge competencies.

## Impact statement

This randomised controlled trial investigates knowledge-based learning outcomes among non-specialist providers following digital training on an evidence-based youth mental health intervention (problem-solving therapy). We compared two digital training formats (self-guided digital training vs. digital training with coaching) and found that both formats led to increased knowledge competency scores, with an incremental effect observed in the coaching arm. We also found higher levels of engagement among participants in the coaching arm. The findings suggest that automated pre-recorded training augmented by periodic coaching is a promising approach that could be used at scale to develop the knowledge base of prospective practitioners of psychosocial interventions in task-sharing initiatives.

## Introduction

Task-sharing of psychotherapies is an effective strategy for improving access to evidence-based mental health care, particularly in low-resource contexts (Naslund et al., 2019b). Scaling this approach requires the expansion of service delivery roles to include a wide range

of non-specialist providers such as lay people and community health workers (Hoeft et al., 2018; Raviola et al., 2019). While digital innovations have been developed and tested with the goal of increasing access to effective task-sharing interventions (Singla et al., 2017; Michelson et al., 2020), these innovations have typically addressed the mode and setting of intervention delivery (e.g., using internet-enabled devices as a vehicle for delivering brief psychotherapies outside of conventional clinic settings). Much less research has been done to evaluate the use of digital technologies for building workforce capacity (Naslund et al., 2019a). This evidence gap is a major barrier to scaling up task-sharing of psychotherapies, given that traditional models of in-person, expert-led training are time- and labour-intensive (van Ginneken et al., 2021; Philippe et al., 2022).

The Premium for Adolescents (PRIDE) programme is a recent exemplar of task-sharing in the field of adolescent mental health. PRIDE was implemented in India from 2016 to 2022 and aimed to address the scarcity of evidence-based interventions for common adolescent mental health problems in the global health context. The goal was to develop and evaluate a suite of scalable, transdiagnostic psychological interventions that could be delivered by non-specialist providers for a variety of mental health presentations in school settings. The programme was intended to generate policy-relevant knowledge in response to India's national initiative for adolescent health, Rashtriya Kishor Swasthya Karyakram. This national policy programme emphasised mental health as a public health priority and schools as an important platform for youth-focused psychosocial interventions (Roy et al., 2019).

PRIDE sought to overcome the resource limitations of expert-led, in-person training by developing a digital learning platform to train non-specialist providers in an evidence-based problem-solving intervention. This learning platform was originally created by Sangath to train non-specialist providers in a brief psychotherapy for adults with depression (Khan et al., 2020) It is designed to host modules comprising video lectures with accompanying role-play demonstrations, narrated teaching slides, self-assessment quizzes, and assigned readings. The Sangath learning platform has recently completed evaluation in a 3-arm randomised control trial (Muke et al., 2020) which compared self-guided digital training and digital training augmented by coaching with the gold standard of in-person, expert-led training.

Building on this body of research, we aimed to evaluate participant engagement and learning outcomes for a modular, digital training course built around a brief transdiagnostic problem-solving intervention for common adolescent mental health problems (i.e., anxiety, depression and conduct difficulties). Our group has previously demonstrated the short- and medium-term, effectiveness of this problem-solving intervention when delivered by lay counsellors in schools serving low-income communities in New Delhi (Michelson et al., 2020; Malik et al., 2021). The goals of the current study were to:

(1) evaluate the effects of digital training on knowledge-based competencies in relation to problem-solving therapy for common adolescent mental health problems;
(2) evaluate the incremental effect of digital training with coaching (DT-C) in comparison with self-guided digital training (DT) on competencies; and
(3) assess participant engagement in, and satisfaction with, the two training conditions.

Our hypotheses were:

(1) participation in either digital training format will lead to increased knowledge-based competency scores among non-specialists; and
(2) DT-C will be more effective than DT at increasing knowledge-based competency scores.

## Methods

### Design and setting

The study was a parallel, two-arm, individually randomised controlled trial design (comparing DT and DT-C) nested within a pre-post intervention study (comparing pre- and post-training learning outcomes for participants across both training conditions).

### Participants

To increase the generalisability of findings, the study sample was drawn from varied backgrounds in India. Participants comprised two groups: (i) university students currently enrolled in a bachelor's-level degree programme in psychology, education or allied fields; and (ii) non-governmental organisation (NGO) staff working as teachers, social workers or mental health advocates. Group (i) was recruited from two co-educational private (one being charity-aided) colleges in Delhi-NCR region; one co-educational private college in Bangalore, Karnataka region; and one girls-only government-aided private college in Mumbai, Maharashtra region. Group (ii) was recruited from four NGOs based in Delhi and one NGO based in Mumbai. Eligible participants in both groups were aged 18 years or older; fluent in written and spoken Hindi or English; and able to access an internet-enabled device as needed to engage in the training. We excluded individuals with prior training in/experience of delivering structured psychotherapies for young people or any other population.

### Sample size calculation

We aimed to recruit 262 participants in the study, with the expectation that 210 participants would complete a follow-up assessment (i.e., allowing for 20% drop-out). For the first hypothesis, this sample size provides 80% power to detect an effect size of 0.19 (i.e., a standardised mean difference (SMD) of post- vs. pre-training scores for 210 participants) at the 2-sided 5% type-I error rate. This indicative effect size (SMD = 0.19) was informed by a systematic review and meta-analysis of online learning evaluations which compared analogous learning conditions (Means et al., 2009). For the second hypothesis, a sample size of $N = 210$ (105 per arm) provides 80% power to detect an effect size (SMD) of 0.39 between the DT and DT-C arms. Due to the enrolment of participants in weekly batches, the final recruited sample size ($N = 277$) slightly exceeded the original target. Data collection was completed in August 2022.

### Participant enrolment

We held online webinars with the collaborating institutions to raise awareness about the study and associated digital training. Webinars were publicised using existing email lists and WhatsApp groups maintained by the various universities and NGOs. The webinars were hosted on Zoom and facilitated by a member of the research

team using a slide show with video demonstration of the digital training course followed by a question-and-answer session with the attendees. Webinars were conducted at regular intervals from March to June 2022 to maintain a rolling flow of referrals. Following the webinar, interested participants were provided with a weblink to the study website (hosted on the REDCap platform) where they were prompted through a series of eligibility questions about age, occupation, device access, language proficiency, and prior training/other experience in psychotherapies. They were subsequently provided with further written information about the study and invited to give consent by means of a digital signature on the study website. Upon completion of the digital training course, all participants received a training completion certificate. Additionally, on completion on the digital training course and the post-training outcome measures, all participants received a gift card worth 500 Indian Rupees (approx. US$6) to offset the cost of data incurred in completing the study.

### Randomisation and blinding

Immediately after completing the baseline questionnaire, participants were randomly allocated to one of the two trial arms. Randomisation was based on a computer-generated list of block sizes 4 and 6 stratified by organisation (NGO or university). This list was programmed into REDCap for automated randomisation. Participants were informed of their allocation by email, which also included a link to access the training programme and login details. Only the data manager (JEJ) had access to the randomisation list and all other study members were blinded to the allocation until final analysis. The participants and coaches were not blinded to allocation status.

### Interventions

#### Self-guided DT arm

The digital training programme contained 16 modules, organised sequentially in two sections: non-specific counselling skills and skills that are specific to problem-solving therapy. The course content was adapted from an existing intervention manual (PRIDE, Sangath, & New Delhi, 2022), which was previously tested in an RCT ($N$ = 250) that compared counsellor-led problem-solving (supported by problem-solving booklets) with problem-solving booklets alone in a target population of school-going adolescents with elevated mental health presentations (Michelson et al., 2020; Malik et al., 2021). The intervention had sustained effects on global psychopathology ($d$ = 0.21), internalising problems ($d$ = 0.22) and idiographic psychosocial problems ($d$ = 0.34) over 12 months. These durable effects were obtained despite a rapid delivery schedule comprising only 4–5 face-to-face sessions (lasting 20–30 min each) over 3 weeks.

The steps taken to translate the manual into a digital curriculum are described in the published study protocol paper (Mathur et al., 2023b). The course was available in two languages, English and Hindi, either of which could be selected by the participants. Participants were expected to progress through the material within 6 weeks of enrolment. The material was accessible in a predetermined sequence, with four modules unlocked each week over four successive weeks. Participants could only progress through the modules in a specified order and had to complete the preceding material before the next set of four modules became accessible. Weekly emails and notifications on the digital platform served as reminders and motivators for course completion. Apart from

addressing technical queries (e.g., related to accessing and navigating the digital platform) through a dedicated WhatsApp number, participants had no other contact with the study team for the duration of the training programme.

#### DT-C arm

In addition to the digital training programme, participants in the DT-C arm received up to four personalised coaching sessions, delivered remotely via voice calls at weekly intervals during the course (average duration of coaching calls = 25 min). In line with the wider pedagogical literature (Irby, 2018), coaching focused on assistive tasks to support individual performance rather than tasks aimed at specific improvements in learning goals. The latter would be more consistent with the related concept of tutoring. Though coaching was primarily delivered through means of phone calls, participants also had the option to ask queries via SMS text messages to their coach in between scheduled coaching sessions. Coaching sessions involved reviewing course concepts, clarifying content-related queries from participants, assisting with time management, troubleshooting other challenges to course completion, and positively reinforcing progress.

There were four coaches (three females; one male), each of whom was a lay counsellor who had previously completed training in the problem-solving intervention; two of these coaches had additionally gained experience of applying the problem-solving intervention in practice. Two of the coaches had previously obtained bachelor's degrees and two had master's degrees. The coaches were also provided with a 2-week training, which consisted of didactic lectures, reading materials, role-play demonstrations, and mock coaching session. Coaches participated in weekly group supervision led by a masters-level Psychologists. Supervision entailed listening to audio-recordings of coaching sessions, which were rated for quality by the coach responsible, their peers and their supervisor (quality rating tool available on request). Ratings covered several aspects of coaching structure and coaching skills, with each item rated from 1–4 (higher scores indicating higher quality). Peers and supervisors also provided formative feedback on recordings and offered suggestions for future coaching sessions, as required. The development and content of the coaching protocol have been described in greater detail in the published study protocol (Mathur et al., 2023b).

### Measures

#### Primary outcome

The primary outcome was the change in scores on a knowledge-based competency measure, the Knowledge Of Problem Solving (KOPS) scale (see Supplementary Material). Taking a broad definition of "competency" as "the extent to which a therapist has the knowledge and skill required to deliver a treatment to the standard needed for it to achieve its expected effects" (Fairburn and Cooper, 2011), the KOPS scale focuses on the former knowledge-based domain. As such, the assessed items correspond to "knowing" and "knowing how" rather than "showing how" or "doing" in the nomenclature of Miller's (1990) hierarchy of clinical competency. Development and validation of the measure have been described elsewhere (Mathur et al., 2023a). The measure comprised five session vignettes for a hypothetical case, with each vignette followed by 3–4 multiple-choice questions that asked about the most appropriate response to a practice-based scenario. Two 17-item parallel forms of the KOPS were administered at baseline and endline assessments, with the sequencing of the two forms

determined at random. Thus, the participants who received version A of the form at baseline received version B of the form at endline and vice versa. A total KOPS score was assigned by summing correct scores (1 point for each correct answer) for 16 items, with one item discarded due to poor psychometric performance. We also conducted a sensitivity analysis using the full 17-item scale.

### Secondary outcomes

*Participants' satisfaction with training*: We used a 26-item version of the eMpowerment, Usefulness, Success, Interest, Caring (MUSIC) questionnaire (Jones and Skaggs, 2016). MUSIC is a measure of satisfaction with educational programmes that have been used in previous digital training trials in India to compare training experiences between groups (Muke et al., 2020; Naslund et al., 2021). Items on the questionnaire were rated on a 6-point scale, covering respective subscale domains of feasibility, acceptability, adoption, and appropriateness. These subscales were scored and analysed separately and not as a total, consistent with prior use. The scores for each subscale ranged from 1 to 6 where higher scores indicate greater levels of satisfaction. Two supplementary free-text items were also used to obtain written qualitative feedback from participants about what they enjoyed the most in the course, as well as suggestions for improvement. These qualitative data have not been reported in the current article.

*Training completion*: This was scored positive for those participants who completed all 16 modules of the digital training.

### Process indicators

Fidelity of coaching sessions was measured in two ways: first, through the number of completed coaching sessions; and second, through the assessed quality of coaching using a new scale developed for the study. Only the ratings provided by supervisors on the quality rating scale (CQRS) described above were considered for quality assessment of the sessions with scores ranging from 1 to 4 (higher scores indicating higher quality).

### Statistical analysis

A statistical analysis plan was finalised before unblinding. Analyses were conducted on an intention to treat principle. Descriptive statistics were used to describe baseline characteristics of participants and variables related to engagement in study procedures (see CONSORT flow diagram, Figure 1). Missing outcomes were imputed using multiple imputations by chained equations under a missing at random assumption. The imputation model was stratified by arm and included the variables in the analysis and those associated with missingness (see Appendix A1). Fifty imputations were performed.

The first hypothesis (analysis of pre vs. post-training competency score) was analysed by fitting a linear regression of the change in competency score between baseline and 6 weeks. The second hypothesis (comparison of DT vs. DT-C) was analysed by fitting a linear regression of the change in competency score between baseline and 6 weeks, testing for a difference between the two arms, adjusted for baseline competency score and strata (NGO vs. university). A similar linear or logistic model was used to compare secondary outcomes (MUSIC subscales and course completion) between DT and DT-C arms, adjusted for strata. We conducted sensitivity analyses for the primary outcome using the 17-item version of the questionnaire, and without imputation (complete-case analysis).

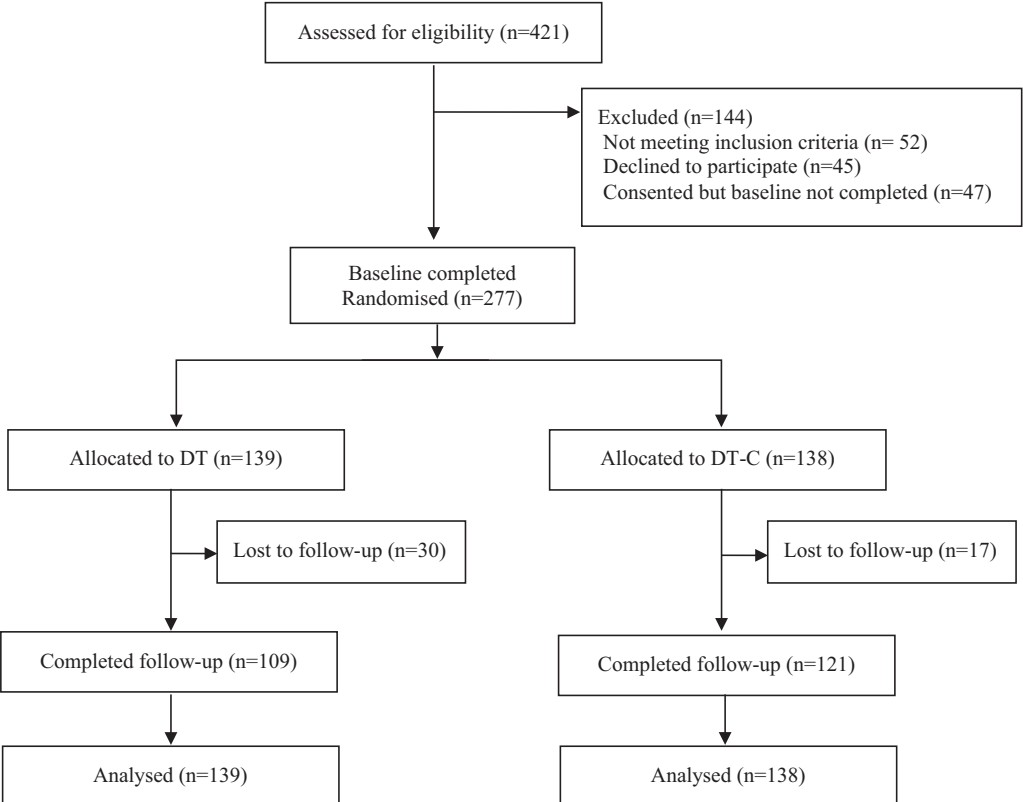

**Figure 1.** Trial flow chart.

We assessed the heterogeneity of training and coaching effects between pre-specified subgroups (age, gender, language, and organisation). For the training effect, we tested for a difference in competency score change between subgroups. For the coaching effect, we tested for an interaction term between trials arm and subgroups. Dose–response effect was investigated descriptively by considering the mean change in competency score by number of DT modules and coaching calls completed.

All analyses were conducted in Stata version 17, and statistical significance was considered at the two-sided 5% alpha level.

## Results

### Participants enrolment and study flow

The collaborating organisations referred 491 individuals, from which 421 (85.8%) were assessed for eligibility (Figure 1). Out of the assessed individuals, 277 (65.8%) enrolled in the study (Figure 1). The mean age of enrolled participants was 26.2 years (SD = 6.8; range: 25.3–27, 95% CI). Most participants were female (229, 82.7%) and NGO members (155, 56.0%) (see Table 1). Participants were randomised to either digital training alone (DT, $n = 139$) or digital training with additional coaching (DT-C, $n = 138$). There was a good balance between the two arms on all baseline characteristics (Table 1). Follow-up at 6 weeks was completed by 230 participants, the rest could not be contacted for follow-up (83.0%). Those lost-to-follow-up tended to be younger, were more likely to be in the DT arm, and were less likely to have completed the digital training (Appendix Table A1). Details of data

completion and outcomes before imputation have been reported in Appendix Table A2.

### Primary outcome

#### Change in knowledge-based competency score after digital training

At baseline the overall mean competency score was 7.01 (range: 0–15, 95% CI 6.62–7.40, $n = 277$; Table 1). At follow-up, the mean score was 8.88 (range: 0–15, 95% CI 8.39–9.38, $n = 230$). Based on imputed data, the mean change in competency score between baseline and follow-up was 1.72 (95% CI 1.33–2.12, $p < 0.001$), corresponding to an effect size (standardised mean difference) of 0.52 (95% CI 0.40 to 0.64).

#### Effect of coaching on knowledge-based competency

Participants randomised to the DT-C arm had greater improvement in competency score compared to those in the DT arm (adjusted mean difference (AMD) adjusted for baseline competency score and stratum = 1.09, 95% CI 0.26–1.92, $p = 0.01$; Table 3), corresponding to an effect size ($d$) of 0.33 (95% CI 0.08–0.58). Results of the sensitivity analyses were similar (Appendix Table A3).

### Intervention completion

Overall, 152 (54.9%) participants completed all 16 modules of the digital training (Table 2), while 43 (15.5%) did not log in even once. Average time to complete the digital training course was 25.7 days (range: 24.0–27.3, 95% CI). Among the 138 participants in the DT-C arm, 69 (50.0%) completed all four coaching sessions, and 23 (16.7%) did not attend any coaching sessions (Table 2).

There was strong evidence that participants in the DT-C arm were more likely to complete the entire digital training (69.6% vs. 40.3%, adjusted odds ratio (OR) = 3.40, 95% CI 2.07–5.60, $p < 0.001$; Table 3). There was some evidence that MUSIC subscale scores were higher in the DT-C versus DT participants (AMD range: 0.11 (Success) to 0.27 (Interest), however the $p$-values ranged from 0.02 (Interest) to 0.25 (Success) with interest being the only significant one; Table 3).

**Table 1.** Baseline characteristics of study participants by arm and combined

| | DT (n = 139) | DT-C (n = 138) | Combined total (n = 277) |
|---|---|---|---|
| Age (years) (mean, SD) | 26.2 (6.8) | 26.2 (7.4) | 26.2 (7.1) |
| Gender (Female) (n, %) | 116 (83.5) | 113 (81.9) | 229 (82.7) |
| Education (n, %) | | | |
| Up to high school | 41 (29.5) | 45 (32.6) | 86 (31.0) |
| University graduate | 98 (70.5) | 93 (67.4) | 191 (69.0) |
| Language of instruction (n, %) | | | |
| Hindi | 106 (76.3) | 102 (73.9) | 208 (75.1) |
| English | 33 (23.7) | 36 (26.1) | 69 (24.9) |
| Recruited from (n, %) | | | |
| NGO | 78 (56.1) | 77 (55.8) | 155 (56.0) |
| University | 61 (43.9) | 61 (44.2) | 122 (44.0) |
| Experience of working with adolescents (n, %) | | | |
| None | 100 (71.9) | 107 (77.5) | 207 (74.7) |
| 1 year or less | 24 (17.3) | 21 (15.2) | 45 (16.2) |
| 2–4 years | 11 (7.9) | 8 (5.8) | 19 (6.9) |
| 5+ years | 4 (2.9) | 2 (1.4) | 6 (2.2) |
| Competency score (mean, SD) | 6.97 (3.37) | 7.04 (3.23) | 7.01 (3.29) |

*Note:* No missing data.
DT, digital training; DT-C, digital training with coaching; NGO, non-governmental organisation; SD, standard deviation.

**Table 2.** Change in competency score for participants by number of completed modules and coaching sessions provided

| | n (%) | Mean change in competency score[a] (SE) |
|---|---|---|
| DT modules completed (n = 277) | | |
| No login | 43 (15.5) | −0.19 (0.65) |
| 0–7 modules | 66 (23.8) | 0.78 (0.43) |
| 8–15 modules | 16 (5.8) | 1.38 (0.80) |
| 16 modules | 152 (54.9) | 2.71 (0.22) |
| Coaching sessions provided (n = 138)[b] | | |
| No sessions | 23 (16.7) | 0.52 (0.89) |
| 1–3 sessions | 46 (33.3) | 2.30 (0.52) |
| 4 sessions | 69 (50.0) | 2.77 (0.34) |

*Note:* Post-hoc $p$-values for chi-squared test for trend are $p < 0.001$ for DT modules completed, and $p = 0.02$ for coaching sessions.
DT, digital training; DT-C, digital training with coaching; SE, standard error.
[a]Mean change in competency score between baseline and follow-up, based on imputed data.
[b]For DT-C arm only.

**Table 3.** Comparison of outcomes at follow-up between DT and DT-C arm (*n* = 277, imputed data)[a]

| | DT (*n* = 139) | DT-C (*n* = 138) | Adjusted[b] mean difference /OR | 95% CI | *p*-value |
|---|---|---|---|---|---|
| | Mean (SE) *n* (%) | | | | |
| Primary outcome | | | | | |
| Change in competency score (mean) | 1.19 (0.29) | 2.26 (0.30) | 1.09 | 0.26, 1.92 | 0.01 |
| Secondary outcomes | | | | | |
| MUSIC – Empowerment (mean) | 4.97 (0.07) | 5.12 (0.07) | 0.15 | −0.06, 0.36 | 0.15 |
| MUSIC – Usefulness (mean) | 5.27 (0.07) | 5.45 (0.06) | 0.18 | 0.00, 0.36 | 0.06 |
| MUSIC – Success (mean) | 4.93 (0.07) | 5.05 (0.07) | 0.11 | −0.08, 0.31 | 0.25 |
| MUSIC – Interest (mean) | 4.90 (0.09) | 5.17 (0.07) | 0.27 | 0.05, 0.49 | 0.02 |
| MUSIC – Caring (mean) | 5.12 (0.06) | 5.26 (0.06) | 0.13 | −0.04, 0.30 | 0.12 |
| Completed all modules (*n* (%)) | 56 (40.3) | 96 (69.6) | 3.40 | 2.07, 5.60 | <0.001 |

CI, confidence interval; DT, digital training; DT-C, digital training with coaching; OR, odds ratio; SE, standard error.
[a]Based on imputed data, except for completed all modules (no missing data). Participants with follow-up competency scores numbered 230 (109 and 121 in DT and DT-C arms, respectively).
[b]All analyses have been adjusted for organisation strata (NGO or university). Change in competency score has been adjusted for baseline competency score.

There were a total of four coaches, most of them female (3, 75%). Their mean age was 29.8 (SD = 8.6) years, two had completed education till bachelors and two had completed masters.

### Process indicators

There was good quality of the coaching session as evaluated by supervisors (CQRS range 3.59 to 3.96, Appendix Table A4).

Overall, participants in the DT-C arm attended 2.8 coaching sessions (SD = 1.54). Those participants who met the course completion criteria attended 3.5 coaching sessions (SD = 0.92), compared with 1.02 coaching sessions (SD = 1.28) for non-completers.

A total of 114 (82.6%) participants in the DT-C arm raised at least one query regarding course content, navigation or technical aspects of which 91 (79.8%) were during the coaching session and the remaining via WhatsApp messages.

### Subgroup analysis

There was some evidence that the increase in competency score after digital training was greater for university students (0.88, *p* = 0.02), participants fluent in English language (0.92, *p* = 0.05), and younger participants whereby older participants showed significantly lower change (−0.72, *p* = 0.07), but not for gender (−0.03, *p* = 0.96) (Table 4). There was no evidence of heterogeneity of the coaching effect by these subgroups (*p*-values for interaction from 0.16 to 0.67; Table 4).

### Dose–response analysis

Table 2 shows the mean competency score according to the number of DT and coaching sessions completed. Participants who did not complete any module showed a reduction in competency score from pre- to post-training (−0.19, SE = 0.65). Those who completed up to half of the modules showed a slight positive increase in competency score (0.78, SE = 0.43). Participants who completed 9–15 modules showed a more substantial increase in competency

(1.38, SE = 0.80), with an even larger positive change observed among participants who completed all 16 modules (2.71, SE = 0.22).

Similarly, participants in the DT-C arm who did not attend any coaching sessions showed a small positive change in competency 0.52 (0.89), whereas those who attended 1–3 coaching sessions showed a relatively larger increase (2.30, SE = 0.52). The largest positive change was seen among those participants who attended all 4 coaching sessions (2.77, SE = 0.34).

### Discussion

This study aimed to evaluate the effects of two digital formats for training non-specialist providers in an evidence-based psychotherapy for common adolescent mental health problems in India. We found that the month-long digital training programme significantly increased knowledge-based competency scores, with the greatest change scores identified in the group who were randomised to receive weekly coaching sessions. Those who received coaching were three times more likely to complete the full training programme compared to participants in the self-directed learning condition. We infer that digital training is a feasible and effective strategy for building the knowledge base of non-specialists involved in initiatives to scale up the task-sharing of psychotherapies and that remotely delivered coaching can optimise learning outcomes further.

To our knowledge, this is the first RCT to investigate learning outcomes among non-specialist providers following digital training in an evidence-based youth mental health intervention in India or any other low- or middle-income country. The findings are consistent with other research on scalable models of educational delivery, which has shown that digital learning platforms can reach high numbers at relatively low cost but may struggle with engagement when used without systems for interpersonal facilitation (Dimeff et al., 2009; Ehrenreich-May et al., 2016; Rakovshik et al., 2016). A related strand of pedagogical research has shown that human interaction can significantly increase engagement with digital educational materials and give rise to better learning outcomes (Reavley et al., 2018).

**Table 4.** Effect-modification of change in competency score and in coaching effectiveness by pre-specified sub-groups

| Sub-group | N | Mean competency score | | Mean change in competency score | Difference between subgroups | 95% CI for difference | *p*-value |
| | | Baseline (n = 277) | Follow-up (n = 277) | | | | |
| --- | --- | --- | --- | --- | --- | --- | --- |
| **Age** | | | | | | | |
| 18–22 | 133 | 8.22 | 10.32 | 2.09 | | | |
| 23+ | 144 | 5.89 | 7.27 | 1.38 | −0.72 | −1.49, 0.05 | 0.07 |
| **Gender** | | | | | | | |
| Male | 48 | 5.85 | 7.60 | 1.74 | | | |
| Female | 229 | 7.25 | 8.97 | 1.71 | −0.03 | −1.09, 1.03 | 0.96 |
| **Language** | | | | | | | |
| Hindi | 208 | 6.78 | 8.27 | 1.50 | | | |
| English | 69 | 7.70 | 10.11 | 2.42 | 0.92 | 0.02, 1.82 | 0.05 |
| **Organisation** | | | | | | | |
| NGO | 155 | 6.83 | 8.17 | 1.34 | | | |
| University | 122 | 7.23 | 9.45 | 2.22 | 0.88 | 0.12, 1.64 | 0.02 |
| Sub-group | N | Mean change in competency score | | Difference between arms[a] | Difference between subgroups[a] | 95% CI for difference | *p*-value |
| | | DT arm (N = 139) | DT-C arm (N = 138) | | | | |
| **Age** | | | | | | | |
| 18–22 | 133 | 1.21 | 3.00 | 1.62 | | | |
| 23+ | 144 | 1.18 | 1.58 | 0.62 | −1.00 | −2.49, 0.50 | 0.19 |
| **Gender** | | | | | | | |
| Male | 48 | 1.77 | 1.73 | 0.35 | | | |
| Female | 229 | 1.08 | 2.38 | 1.25 | 0.90 | −1.04, 2.84 | 0.36 |
| **Language** | | | | | | | |
| Hindi | 208 | 0.87 | 2.15 | 1.36 | | | |
| English | 69 | 2.24 | 2.58 | 0.16 | −1.20 | −2.88, 0.47 | 0.16 |
| **Organisation** | | | | | | | |
| NGO | 155 | 0.83 | 1.85 | 0.94 | | | |
| University | 122 | 1.65 | 2.78 | 1.27 | 0.32 | −1.19, 1.84 | 0.67 |

*Note:* Based on imputed data (except baseline competency score, no missing). Participants with follow-up competency scores numbered 230 (109 and 121 in DT and DT-C arms, respectively). CI, confidence interval; DT, digital training; DT-C, digital training with coaching; NGO, non-governmental organisation.
[a]Adjusted for strata and baseline competency score.

The overall completion rate of 54.9% should be considered in the context of a voluntary training programme where there were no incentives for participation other than a certificate of completion. Such conditions are well known to be associated with high levels of attrition in "Massive Open Online Courses" (MOOCs) and other open-access courses, where completion rates typically cluster around 5–10% (Allione and Stein, 2016; Badali et al., 2022). Against this low benchmark, the observed completion rate appears to be relatively encouraging. A higher completion rate is conceivable under alternative conditions where motivation could be enhanced through formalised academic credit or a clear-cut trajectory from training to practical implementation/qualified practitioner status.

It is likely – and consistent with the dose–response analysis – that the greater knowledge demonstrated by participants in the coaching condition was related to more extensive engagement with the programmed content. The deployment of coaches potentially limits the scalability of digitally delivered training, not least as most existing models for coaching have utilised experts (Frank et al., 2020). In contrast, the coaches in our study were non-specialist providers themselves who did not have professional qualifications or substantive training, further adding to scalability of the digital platform.

Although the pre-post changes were significant overall and the effect sizes moderate to large, in absolute terms participants were able to answer just 1 or 2 additional questions correctly after the training. Even in the coaching condition, the post-training mean score of 9.30 corresponded to approximately 7 incorrect answers out of 16 questions (43.8%). Hence, there is clearly a need for further learning support, such as supervised practice. The relatively small change in competency scores may also reflect motivational issues in the sample, given that none of our study participants were enrolled in practice-based courses or employed in practice roles

that would necessarily facilitate real-world applications. Different results may have been obtained for a more selected service-oriented sample who were expecting to apply the training directly into practice.

Another limitation of our study concerns the use of a knowledge-based competency measure, rather than a measure of demonstrated skills. That said, our competency measure was validated in the study context and consisted of counselling vignettes that approximated real-life situations. This emphasis on applied knowledge ("knowing how") rather than purely theoretical understanding strengthens ecological validity, though we accept that it cannot substitute entirely for a gold-standard observational assessment of clinical skills. For example, the observer-rated ENhancing Assessment of Common Therapeutic factors (ENACT) scale was designed for training and supervision of non-specialist providers of psychological interventions in culturally diverse and resource-constrained settings (Kohrt et al., 2015). However, we note that ENACT functions as a measure of common factors in psychotherapies (i.e., competencies that are implicated in the effective delivery of any psychotherapy) and does not assess competencies that are unique to problem-solving or other discrete practice elements. ENACT's broad-based assessment of therapeutic skills, supplemented with observer-rated items covering more specific therapeutic skills, would ideally be deployed after a period of case-based practice, rather than following a didactic training of the type used in the current study. A further limitation is that we did not assess the prospective impact of training on clinical outcomes. However, other research has shown that higher post-training knowledge is associated with better mental health outcomes for treated cases (Rakovshik et al., 2016; Milligan-Saville et al., 2017) and knowledge could be considered as a pre-requisite for effective transfer to practice.

In conclusion, digital training is a promising strategy, especially when supplemented by remote coaching, for growing the workforce needed to deliver evidence-based psychotherapies at scale. Importantly, such trainings involve a one-time investment of expert resources in designing the curriculum after which there is a comparably much smaller cost for implementation. Thus, the shift towards automated, pre-recorded training offers a substantial scalability advantage over conventional expert-led workshops, which must be repeated in real-time to successive cohorts. Large-scale digital programmes with relatively low running costs could be used to select promising candidates for more resource-intensive further training and supervised practice.

Future research should examine how these knowledge-based competencies can be translated into actual therapy skills, for example through supervised case-based practice, and directly address questions about how to sustain training benefits over time. Research is also needed to establish the generalisability of digital training formats for other psychosocial interventions and in diverse contexts, ultimately serving to scale up task-sharing initiatives aimed at reducing the mental health care gap globally.

**Open peer review.** To view the open peer review materials for this article, please visit http://doi.org/10.1017/gmh.2023.81.

**Supplementary material.** The supplementary material for this article can be found at https://doi.org/10.1017/gmh.2023.81.

**Data availability statement.** Anonymised participant data, data dictionary and case report forms will be made available on *datacompass.lshtm.ac.uk* by 12 months after trial completion. Data will be shared after approval by the corresponding author, following a reasonable submitted request. The study protocol and statistical analysis plan are publicly available on *clinicaltrials.gov*.

**Acknowledgements.** We are grateful to members of the PRIDE Scientific Advisory Group, chaired by Prof. Chris Fairburn, and colleagues who contributed to intervention and research activities across the various phases of the study, including Shreyas Kamat, Abhija Teli, Keshav Shakya, Niket Agarwal, Rukhsar Akram, Sai Priya Kumar, Aarohi Sharma, Manogya Sahay, Meenakshi Pandey, Sarita Rao, Shouaib Ahmed and Himanshu Gupta. We also gratefully acknowledge the contributions made by the Population Foundation of India in designing the digital training course.

**Author contribution.** S.M., D.M. and V.P. led on the development of the digital training platform and coaching protocol, with contributions from T.S. and K.M. S.M. led on drafting of the manuscript, with critical inputs from D.M. and V.P. H.A.W., M.N., B.L. and A.P.F. drafted the statistical analysis plan. S.M., T.S., J.E.J., P.N. and R.M. contributed to the coordination and implementation of the trial. All authors read and approved the final manuscript. D.M. and V.P. contributed equally as senior authors.

**Financial support.** This study was supported by a Principal Research Fellowship awarded to V.P. by the Wellcome Trust (grant number 106919/A/15/Z). The funders had no role in the design of the study, data collection or the writing of the manuscript. For the purpose of open access, the author has applied a CC BY public copyright licence to any Author Accepted Manuscript version arising from this submission.

**Competing interest.** The authors declare no conflict of interest.

**Ethics approval.** Institutional Review Board approvals were obtained from Sangath (the implementing organisation in India); Harvard Medical School, USA (the sponsor) and the London School of Hygiene and Tropical Medicine, UK (a collaborating institute).

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

## Appendix – Additional Results

In addition to the primary and secondary outcomes to be imputed (change in competency score and MUSIC subscales), the imputation model included the following complete variables: age, gender, language, education, current occupation, organization, and number of DT module completed. Imputation was conducted stratified by arm.

Analysis models:

1. Primary analysis: based on 16-item competency score and multiply imputed data, see main paper for full details.
2. 17-items competency score: Using the competency score including all 17 items, before exclusion of 1 item with low discrimination. Based on multiply imputed data, using same imputation model than primary analysis.
3. Complete cases: Based on 230 participants who completed the competency score at follow-up.

**Table A1.** Characteristics of participants who did and did not complete follow-up

| | Completed follow-up (n = 230) | Did not complete (n = 47) | p-value |
|---|---|---|---|
| **Study arm** (n [%]) | | | |
| DT | 109 (47.4) | 30 (63.8) | 0.04 |
| DT-C | 121 (52.6) | 17 (36.2) | |
| **Age (years)** (mean [SD]) | 26.52 (7.47) | 24.43 (4.66) | 0.07 |
| **Gender** (Female) (n [%]) | 189 (82.2) | 40 (85.1) | 0.63 |
| **Education** (n [%]) | | | |
| Up to 12th | 74 (32.2) | 12 (25.5) | 0.37 |
| Graduate | 156 (67.8) | 35 (74.5) | |
| **Language** (n [%]) | | | |
| Hindi | 106 (76.3) | 102 (73.9) | 0.17 |
| English | 33 (23.7) | 36 (26.1) | |
| **Occupation** (n [%]) | | | |
| Student | 144 (62.6) | 36 (76.6) | 0.001 |
| Counsellor | 37 (16.1) | 2 (4.3) | |
| Teacher | 16 (7.0) | 4 (8.5) | |
| Social Worker | 25 (10.9) | 0 (0.0) | |
| Mental Health Advocate | 4 (1.7) | 0 (0.0) | |
| Other | 4 (1.7) | 5 (10.6) | |
| **Organisation** (n [%]) | | | |
| NGO | 132 (57.4) | 23 (48.9) | 0.29 |
| University | 98 (42.6) | 24 (51.1) | |
| **Experience of Mental Health Work** (n [%]) | | | |
| None | 170 (73.9) | 37 (78.7) | 0.94 |
| 1 year or less | 39 (17.0) | 6 (12.8) | |
| 2–4 years | 16 (7.0) | 3 (6.4) | |
| 5+ years | 5 (2.2) | 1 (2.1) | |
| **Organization code**[**] | | | |
| 1 | 25 (10.9) | 0 (0.0) | 0.001 |
| 2 | 20 (8.7) | 1 (2.1) | |
| 3 | 26 (11.3) | 9 (19.1) | |
| 4 | 26 (11.3) | 11 (23.4) | |
| 5 | 27 (11.7) | 3 (6.4) | |
| 6 | 57 (24.8) | 20 (42.6) | |
| 7 | 30 (13.0) | 2 (4.3) | |
| 8 | 19 (8.3) | 1 (2.1) | |
| **DT modules completed** (n = 277) | | | |
| No login | 22 (9.6) | 21 (44.7) | <0.001 |
| 0–7 modules | 43 (18.7) | 23 (48.9) | |
| 8–15 modules | 15 (6.5) | 1 (2.1) | |
| 16 modules | 150 (65.2) | 2 (4.3) | |
| **Coaching sessions completed** (n = 138)[*] | | | |
| No sessions | 13 (10.7) | 10 (58.8) | <0.001 |
| 1–3 sessions | 39 (32.2) | 7 (41.2) | |

(*Continued*)

**Table A1.** (*Continued*)

| | Completed follow-up (*n* = 230) | Did not complete (*n* = 47) | *p*-value |
|---|---|---|---|
| 4 sessions | 69 (57.0) | 0 (0.0) | |
| **Baseline competency score** (mean, SD) | 6.93 (3.22) | 7.33 (3.61) | 0.11 |

*DT-arm only. P-values calculated with Pearson's chi-squared tests (arm, gender, education, language, type of organization) or Fisher's exact tests (occupation, experience, organization, DT modules completed, coaching sessions completed) for categorical variables and t-tests for continuous variables (age, baseline competency score).
**Smallest site with 2 participants regrouped with a larger similar site.

**Table A2.** Observed competency and MUSIC outcomes at follow-up, by arm and overall

| | DT | DT (*n* = 139) | DT-C | DT-C (*n* = 138) | | Overall (*n* = 277) |
|---|---|---|---|---|---|---|
| | *n* | Mean (SD) | *n* | Mean (SD) | *n* | Mean (SD) |
| **Competency score after 6 weeks** | 109 | 8.30 (3.79) | 121 | 9.40 (3.75) | 230 | 8.88 (3.80) |
| **Change in competency score** | 109 | 1.51 (2.82) | 121 | 2.37 (2.99) | 230 | 1.96 (2.94) |
| **MUSIC – Empowerment** | 105 | 5.03 (0.62) | 121 | 5.14 (0.73) | 226 | 5.09 (0.68) |
| **MUSIC – Usefulness** | 105 | 5.34 (0.61) | 121 | 5.46 (0.64) | 226 | 5.40 (0.63) |
| **MUSIC – Success** | 105 | 4.97 (0.63) | 121 | 5.06 (0.68) | 226 | 5.02 (0.66) |
| **MUSIC – Interest** | 105 | 4.97 (0.72) | 121 | 5.21 (0.69) | 226 | 5.10 (0.71) |
| **MUSIC – Caring** | 105 | 5.17 (0.54) | 121 | 5.24 (0.65) | 226 | 5.21 (0.60) |

DT = Digital Training; DT-C = Digital Training and Coaching; SD = Standard Deviation

**Table A3.** Sensitivity analyses

| Change in competency score after digital training | | | | | | |
|---|---|---|---|---|---|---|
| | | Mean competency score | | | | |
| Analysis | *N* | Baseline | Follow-up | Mean change | 95% CI | *p*-value |
| Primary analysis model | 277 | 7.01 | 8.73 | 1.72 | (1.33, 2.12) | <0.001 |
| 17-items competency score | 277 | 7.32 | 9.22 | 1.74 | (1.34, 2.14) | <0.001 |
| Complete cases | 230 | 6.92 | 8.88 | 1.97 | (1.58, 2.35) | <0.001 |
| **Effect of coaching on competency** | | | | | | |
| | | Mean change in competency score | | | | |
| Analysis | *N* | DT (*n* = 139) | DT-C (*n* = 138) | Adjusted* mean difference | 95% CI | *p*-value |
| Primary analysis model | 277 | 1.19 | 2.26 | 1.09 | (0.26, 1.92) | 0.01 |
| 17-items competency score | 277 | 1.25 | 2.23 | 1.00 | (0.16, 1.83) | 0.02 |
| Complete cases | 230 | 1.51 | 2.37 | 0.91 | (0.19, 1.63) | 0.01 |

DT = Digital Training; DT-C = Digital Training and Coaching
*Mean difference between DT and DT-C arm, adjusted for organization strata and competency score at baseline.

**Table A4.** Coach characteristics and Quality rating of coaching sessions

|  | Coaches (*n* = 4) |
|---|---|
| **Gender** (Female) (*n* [%]) | 3 (75) |
| **Age (years)** (mean [SD]) | 29.8 (8.6) |
| **Education** (*n* [%]) |  |
| Bachelor | 2 (50) |
| Masters | 2 (50) |
| **Training** (*n* [%]) |  |
| Problem-Solving Counselling (F2F) and Coaching | 2 (50) |
| Problem-Solving Counselling (Digital) and Coaching | 2 (50) |
| **Participants coached** (range) | 29-40 |
| **Coach average CQRS score**\* (range) | 3.59-3.96 |

*CQRS = Coaching Quality Rating Scale (developed for the current study). 25 assessment points, with overall score ranging from 1 (being limited) to 4 (advanced) fidelity score.