## [Reviewer Report]

Prof Judy Bass and Dr Dixon Chibanda

Editor-in-Chief, Global Mental Health

7 June 2023

Dear Prof Bass and Dr Chibanda,

RE: Manuscript submission - “Developing psychotherapeutic competencies in non-specialist providers: a pre-post study with a nested randomised controlled trial of a coach-supported versus self-guided digital training course for a problem-solving psychological intervention in India”

I am pleased to submit the above-named manuscript for consideration as an original research paper in Global Mental Health. I can confirm the manuscript is not under review with any other journal and we have no conflicts of interest. The trial has been pre-registered at clinicaltrials.gov (NCT05290142).

The submitted paper originates from a larger six-year programme of research called PRIDE (PRemIum for aDolEscents), which aims to develop and scale up a suite of school-based, transdiagnostic psychosocial interventions for adolescents with common mental health problems in India. Previous PRIDE studies, including a mixed-methods evaluation study published in Global Mental Health (link here), have produced a raft of evidence on the performance of linked intervention components organised around a stepped care architecture. 

The submitted research paper reports the findings of a complementary evaluation of an innovative digital training course, which is intended to up-skill prospective providers for PRIDE’s first-line problem-solving intervention. The study incorporated a pre-and-post study design with a nested RCT, evaluating digital training delivered in a self-guided format or with coaching. This is the first randomised controlled trial to investigate learning outcomes among non-specialist providers following digital training in an evidence-based youth mental health intervention. We found that digital training significantly increased non-specialists’ competency to deliver the problem-solving intervention, especially when aided by remote coaching which also led to higher engagement in the training. This shift towards automated, pre-recorded training offers a substantial scalability advantage over conventional expert-led workshops is a promising and relatively low running cost alternative. 

These findings will be used to support wider efforts to scale up evidence-based mental health interventions for young people. We expect that the paper will be of significant interest to researchers and practitioners in the field of community (youth) mental health, and particularly to those working in low-resource settings.

Thank you in advance for your consideration.

Yours sincerely,

Vikram Patel

The Pershing Square Professor of Global Health and Wellcome Trust Principal Research Fellow, Harvard Medical School

Professor, Harvard T.H. Chan School of Public Health

E: vikram:patel@hms.harvard.edu

T: +91-9822132038

---

## [Reviewer Report]

Thank you for the invitation to review this interesting manuscript describing an innovative trial aimed at evaluating and comparing a coach-supported and self-guided digital training course for a problem-solving psychological intervention in India. The study methods appear rigorous, and the findings are novel, demonstrating the potential effectiveness of digital training methods in increasing participant knowledge, and the added impact of the coaching on participant engagement, training completion and outcomes. See few suggestions for consideration:

First, additional details are needed about the KOPS primary outcome measure of competency, including how this measure captures clinical competencies or skills, and how it was validated. Given that this is a written self-administered test, it seems like it would be best described as a knowledge assessment. The limitations with the measure are acknowledged in the discussion, though the discussion should be expanded to weigh the pros/cons of such a measure with existing gold-standard approaches for competency assessment (i.e., such as the ENACT scale and ratings).

Second, additional details are needed about the nature of the problem-solving therapy covered in the 2 training programs, and evidence of its effectiveness. The program manual is cited, but clinical impact of this therapy should be summarized in the text. The study also mentions that participants could be Hindi or English speaking. Clarification is needed about the language of the digital training, assuming it was available in both languages.

Third, it is not clear how to interpret the differences in scores in the MUSIC model questionnaire between groups. It does not appear that this questionnaire has been previously used for comparing groups in digital training studies. Some additional details and description of the questionnaire and how to interpret the scores would be helpful.

Fourth, the low engagement among participants across both conditions requires greater attention and reflection in the discussion. While it is impressive that the addition of the coaching substantially improved engagement and outcomes, there are still a large number of participants who did not complete the training, or did not start the training at all. It would be helpful to consider how this represents a challenge for informing the scale up of the training and task-sharing efforts more broadly, and also consider what strategies could be employed to potentially overcome low engagement among target learner groups.

Lastly, the findings are currently overstated, because it is not possible to conclude that digital training was ‘effective’ for growing the workforce or scaling up the task-sharing or psychotherapies, unless there is data on whether any of the participants trained in this study went on to deliver care? Instead, recommend rewording throughout that this is a ‘potentially effective’ strategy, or promising strategy, and that the effectiveness is demonstrated in increasing knowledge (rather than scaling up psychotherapies).

---

## [Reviewer Report]

The study design is excellent; however the conclusions drawn are not supported by the study. As noted in the limitations, this relates to knowledge and knowledge acquisition pre- and post- course, which is not the same as the transfer of learning to practice. Thus, the conclusions linking this educational design to enhancing access to psychotherapy and task-sharing are not supported by the study.

Early in the introduction, there is reference to knowledge-based competency, and the improvement of competency scores. Whilst there are many, many definitions of competency around, these commonly relate to practice and performance, and the integration of knowledge alongside skills, attitudes and behaviours, to practice - rather than the component knowledge, skills, attitudes or behaviours. Here are a couple of key papers that you may wish to refer to: https://human-resources-health.biomedcentral.com/articles/10.1186/s12960-019-0443-8 and https://www.ncbi.nlm.nih.gov/pmc/articles/PMC4054715/. A useful and commonly cited reference when categorizing learning outcomes is Miller’s pyramid. My interpretation of this study is that it targeted knows and knows how, only’; and thus does not address competency development (shows how and does levels). Unless the study is extended to assess competencies and transfer to practice, my first recommendation is that this paper refers only to the increase in knowledge and knowledge retention as measured pre-test and post-test, not competency. At this stage you may wish to cite literature that links knowledge and knows-how with transfer to practice – but competence (performance) was not targeted during the learning programme or the coaching.

The approach to coaching also seemed to be outside of my interpretation of coaching, which would relate more to practice experience and transfer from learning to practice; instead it resembled tutoring. Thus the comparison is between self-paced study, and self-paced study with access to a course tutor. I also query the quality of training and support provided to the ‘coaches’ – the manuscript refers to the quality of coaching being scored, but there is no reference to providing feedback or support to the coaches. If this was part of the design, this would be important to mention. My second recommendation thus relates to the language used to describe the role of the ‘coach’.

My third reflection may not be relevant if the first recommendation is taken into account, and relates to the discussion around task-sharing. The introduction describes task-sharing in a way that may be misinterpreted to mean that tasks and healthcare services that are traditionally part of regulated scopes of practice may now be provided by lay people after completing a knowledge-based short course. I’m sure that is not the intention of the authors, and the misperception that this is aiming to ‘dumb down’ service provision could be prevented if for example, example tasks are provided, or the discussion relates to role optimization, and the steps taken to ensure that the quality of services received is not affected by the distribution of tasks amongst the team. For example, one argument may be that through enhancing the training in psychotherapies of lay people and community health workers, a first level access to psychotherapy and initial support can be provided which increases access to trained health workers, who can then refer for more specialized care where needed.

There is merit in revising the manuscript – it is an interesting finding that self-paced learning can be enhanced with the support of a tutor. However, the conclusions should reflect what is being assessed and the increase in knowledge, with perhaps recommendations for supported/coached transfer to practice. In relation to the task-sharing discussion, it may be that the conclusions find that certain tasks can be learned through self-paced learning, rather than the blanket recommendation that self-paced supports task-sharing, but instead to reflect on self-paced plus a tutor as a more effective learning intervention than self-paced learning alone.

---

## [Reviewer Report]

Thank you for resubmitting this revised manuscript. The authors have thoughtfully responded to my comments, and the revised manuscript is significantly improved. This is an interesting study and important contribution to the global mental health literature on efforts to train the non-specialist workforce. I have no further comments at this time.

---

## [Reviewer Report]

Thank you for submitting the revisions, they have clarified the reviewers concerns. To finalize the manuscript, we request a few additional minor revisions:

1. Update the title to match better reflect the knowledge competences focus of the intervention (rather than a digital training course for a problem solving intervention).

2. Update the abstract, including the initial sentence, to similarly reflect that the focus was not on training of the actual problem solving intervention as well as edits that reflect the other adjustments made to reviewer comments (line 12 - specifying knowledge competencies).

3. Update the impact statement as the digital training was not on the evidence-based youth mental health intervention (problem-solving therapy) but rather than knowledge content.

4. Update the keywords based on changes to the manuscript.

---

## [Reviewer Report]

14th November 2023

Dear Prof, Bass,

Re: Revised manuscript, “Developing knowledge-based psychotherapeutic competencies in non-specialist providers: a pre-post study with a nested randomised controlled trial of a coach-supported versus self-guided digital training course for a problem-solving psychological intervention in India”

We are pleased by the positive response to our first round of revisions. We have made further minor changes as requested. 

In so doing, we have responded to each of your comments in turn. 

We trust that these revisions prove satisfactory and look forward to your response in due course.

Yours sincerely,

Prof. Vikram Patel

1.Update the title to match better reflect the knowledge competences focus of the intervention (rather than a digital training course for a problem solving intervention).

Title has been updated to the following, 

“Developing knowledge-based psychotherapeutic competencies in non-specialist providers: a pre-post study with a nested randomised controlled trial of a coach-supported versus self-guided digital training course for a problem-solving psychological intervention in India”

2. Update the abstract, including the initial sentence, to similarly reflect that the focus was not on training of the actual problem solving intervention as well as edits that reflect the other adjustments made to reviewer comments (line 12 - specifying knowledge competencies).

The first line in the abstract has been modified and reproduced below for ease of reference, 

“We evaluated a digital learning programme for non-specialists, with the goal to develop knowledge-based competencies related to a problem-solving for adolescents.”

3. Update the impact statement as the digital training was not on the evidence-based youth mental health intervention (problem-solving therapy) but rather than knowledge content.

The first line in the impact statement has been modified and reproduced below for ease of reference, 

“This randomised controlled trial investigates knowledge-based learning outcomes among non-specialist providers following digital training on an evidence-based youth mental health intervention (problem-solving therapy).” 

4. Update the keywords based on changes to the manuscript.

Keywords have been updated to the following,

“Randomised controlled trial; knowledge-based competency; digital training; capacity building; problem-solving intervention; adolescent mental health; India.”